# Tapping-Actuated Triboelectric Nanogenerator with Surface Charge Density Optimization for Human Motion Energy Harvesting

**DOI:** 10.3390/nano12193271

**Published:** 2022-09-20

**Authors:** Marcos Duque, Gonzalo Murillo

**Affiliations:** Department of Nano and Microsystems, Instituto de Microelectrónica de Barcelona (IMB-CNM, CSIC), 08193 Bellaterra, Spain

**Keywords:** energy harvesting, triboelectricity, triboelectric nanogenerator, TENG, contact-separation mode, corona charging, IoT

## Abstract

In this article, triboelectric effect has been used to harvest mechanical energy from human motion and convert it into electrical energy. To do so, different ways of optimizing the energy generated have been studied through the correct selection of materials, the design of new spacers to improve the contact surface area, and charge injection by high-voltage corona charging to increase the charge density of dielectric materials. Finally, a triboelectric nanogenerator (TENG) has been manufactured, which is capable of collecting the mechanical energy of the force applied by hand tapping and using it to power miniaturized electronic sensors in a self-sufficient and sustainable way. This work shows the theoretical concept and simulations of the proposed TENG device, as well as the experimental work carried out.

## 1. Introduction

The triboelectric effect, also called contact electrification, has been known for thousands of years and implies that when two different materials come into contact, charges will be transferred from one material to the other, depending on their respective electron affinity [1]. Due to its inherent characteristics, the triboelectric effect can cause extremely high voltages. Traditionally, it has been considered a negative effect, especially in the electronics industry, due to the damage that it can cause to integrated circuits. In 2012, triboelectric nanogenerators (TENGs) were invented to take advantage of this previously negative effect and use it for energy harvesting [2,3].

Due to the great advancement of technology and telecommunications, the interest in the Internet of things (IoT) is increasing rapidly [4]. This concept allows the interconnection of thousands of wireless sensors to capture environmental information and make decisions without human interaction. Nevertheless, these sensors must be powered, and this is a major drawback due to their dependence on batteries. An alternative to avoid the use of batteries is to harvest ambient energy and convert it into electricity.

A lot of attention has been dedicated to the use of TENGs for energy harvesting, as they have been shown to be an easy and low-cost way of converting mechanical energy into electricity. This mechanical energy is produced by many sources such as ambient vibrations [5,6,7], magnetic fields [8,9], human motion [10,11,12,13,14,15,16,17], wind [18,19,20], water [21], waves [22,23], vibration by moving vehicles [24,25] and time-limited and random vibrations [26,27,28]. This widely distributed and collected ambient mechanical energy can be used to supply low-power electronic sensors in a self-sufficient way and can be used for common IoT applications for predictive maintenance, system sensing, or the measurement of environmental parameters.

As shown in Figure 1, there are four basic working modes to operate a TENG [29]. Contact-separation mode uses polarization in the vertical direction, so the system energy is increased when the electrodes separate, which corresponds to the decrease in the capacitance in a parallel-plate capacitor. Lateral-sliding mode uses polarization in the lateral direction because of the relative slip between the two materials. The material friction creates the charge separation, as well as a capacitance change. Single-electrode mode harvests energy from a freely moving surface without opposite electrode. Free-standing mode is designed for power generation by electrostatic induction between a pair of electrodes, due to the presence of a sliding charge structure. In many cases, two or more modes can work together.

Contact-separation mode is the one used in our work. This mode requires an external force for the triboelectric material and the electrode to come into contact, to later separate the contact surfaces. This subsequent separation is usually carried out by means of springs, a spacer, or the deformation of the material itself. The use of additional spacers, by means of gaskets [30,31], sponges [32,33], and springs [34,35] or the general deformation of the substrate such as an arched shape [36,37] or triangle prisms [38], increases the cost of the TENG due to greater manufacturing complexity. In addition, many spacers require a greater expansion of the deformation area.

Previous works have proposed a TENG with an etched pattern spacer [39]. This design etches different patterns into the substrate and folds them into small spacers that are evenly distributed on the contact surface. This system presents problems in the generation of energy, due to the fact that there is always a contact between the top electrode and the triboelectric material and, thus, a deficiency in the work area contact. Here, in this work, a system using an engraving pattern in the outer zone of the contact surface of the materials is presented.

Voltage and current outputs of TENGs are proportional to the triboelectric charge density on the contact surfaces. Therefore, a key approach to improve the TENG output performance is to increase the triboelectric charge density through the correct choice of utilized materials, e.g., [40,41,42]. That is, by choosing materials with a higher electron transfer capacity after contact electrification, a higher energy generation will be achieved. Surface modification enlarges the surface area, such as: nanoparticles self-assembly [35], pyramid patterns from photolithographic patterning [36], and surface dry-etched polymer nanowires [43,44]. Structure optimization maximizes the contact area, such as: cylindrical rotating TENG with multiple-layer integration [45] or an advanced PCB composite disk-structure TENG with narrow gratings [46] and the use of a specific power management circuit named Bennet’s doubler to maximize energy efficiency [47]. However, further improvements in performance are needed to be able to supply low-power electronic sensors in a self-sufficient and sustainable way.

Due to the intrinsic properties of dielectric materials to almost permanently retain large amounts of charge (known as electret), the surface charge density of dielectric films can be increased several times by charge injection. Charge injection methods include ionized-air injection, plasma polarization, high-voltage corona charging, and electron beam bombardment [48,49]. High voltage corona charging is the simplest, cheapest, and most widely used process in industrial manufacturing.

In this work, research on the triboelectric materials with the highest intrinsic surface charge density and lowest cost is carried out. Furthermore, a study on the increase in charge injection of different triboelectric materials, by means of high-voltage corona charging, is performed. In addition, an adaptation of a triboelectric generator with engraved pattern spacers is proposed for the improvement of the contact area and its subsequent fabrication. In order to validate the measurements obtained after the injection of charge, a finite element modeling (FEM) has been carried out using COMSOL Multiphysics. Finally, an example of an application is shown, where the mechanical energy from hand tapping is collected and used to power a group of 30 light emitting diodes (LEDs).

## 2. Experimental Section

### 2.1. Study of Triboelectric Materials

The selection process of the materials to be studied is based on numerous works reporting triboelectric series [41,42]. For this case, materials with the highest intrinsic surface charge density have been chosen, also taking into account the material cost. These selected materials are polytetrafluoroethylene (PTFE), polyamide (Kapton), polyether ether ketone (PEEK), and biaxially oriented polyethylene terephthalate (BoPET, Mylar).

The open-circuit voltage *V_oc_* of a *TENG* can be expressed as [50]:(1)$$V_{OC}\left(TENG\right)=\frac{\sigma \;x\left(t\right)}{\varepsilon_{0}}\;,$$
where σ is the surface triboelectric charge density between the electrification materials, $\varepsilon_{0}\;$ is the vacuum permittivity and is the separation distance.

For the characterization of these materials, we integrated different triboelectric materials in a test platform (10 cm × 10 cm) made of PMMA with four metal guides and four springs to retract both surfaces after touching. The triboelectric material and the copper layers are disposed of in between the PMMA structure faces. Different samples of the triboelectric material, 5 cm × 5 cm in size and 50 µm thick, have been used. As shown in Figure 2a, two copper electrodes and a single dielectric material are used (contact separation mode). Periodically, the top electrode separates after getting in contact with the dielectric surface to effect charge transfer. In order to carry out the electrical measurements, a characterization setup has been assembled (Figure 2b,c) consisting of a stepper motor (Zaber LSQ075B-T3-MC03 and X-MCB1-KX13B), a dynamometer (Mark M5i and MR03-20 sensor for a maximum force of 100 N), a sourcemeter (Keithley 2470), and a LabVIEW program that controls the entire electrical characterization setup.

For the characterization of the materials, the linear motor produces a vertical motion to make physical contact between the material and the top electrode. A force of 50 N, which is approximately the force generated by a human footstep, is applied. Figure 3 shows the different triboelectric materials and the average generated voltages. As observed, voltages ranging from −35 V to −57 V can be achieved with a single piece of material. All the measurements were performed with three samples of each type (*n* = 3). For all the tests, the measured ambient temperature and relative humidity were around 30 °C and 20%, respectively.

### 2.2. Surface Charging Process

To increase the surface charge density of dielectric films, as already mentioned above, charge injection is performed by high-voltage corona charging. To do so, as shown in Figure 4a, a PCB with multiple metal tips was designed and manufactured. Using a high-voltage source (Frederiksen 3670.60), a high voltage is applied to the multi-tip electrode. The current flows from the high-potential multi-tip electrode to the ground plane, through the air, by ionizing and creating a region of plasma around the needles. The ions eventually pass the charge to lower potential areas of the dielectric material. In order to clarify this process, Figure 4b shows the schematic of the injection process discussed above.

Firstly, to study the optimum voltage and time to be applied to the corona, PTFE has been chosen as the material, since it is the material that generates the highest output voltage for an applied force. A voltage sweep is performed from 1000 V to 6000 V, with a gradual increase of 1000 V. Figure 5a shows the open-circuit voltage that the material can generate with an electrode separation of 5 mm. The greater the voltage applied to the corona, the greater the increase in charge density of the material.

For a constant voltage of 6000 V applied to the corona discharge, we performed a study on how long it takes to achieve the maximum surface charge. As can be seen in Figure 5b, after 15 min of application time, the maximum voltage generated stabilizes, and, therefore, the maximum surface charge density that the material can support is reached.

Although the humidity and temperature were monitored during the charging process, it can be very sensitive to tiny changes in these parameters, air pressure, or surface cleanness. All the parameters and samples were identical; however, a variability can be observed from one to another. The best future solution to increase reproducibility is to have a specific chamber with a specific gas at controlled pressure and temperature.

Finally, the charge injection is carried out by applying a constant voltage of 6000 V and a period of 15 min. In the Figure 6a, it can be seen how the materials with the lowest intrinsic charge density, such as PEEK and Mylar, are now the ones with the highest surface charge density and, therefore, the ones that generate the highest maximum voltage.

Concerning the permanent effect of this polarization, the surface charge density is limited by the breakdown of the electric field in the air. This ion-injection method is an effective way to increase the output power by up to 25 times. This has been proven to be stable over 5 months and 400,000 continuous operation cycles [51]. In our case, as shown in Figure 6b, after 1500 cycles, the voltage drops 6%, and, after 2500 cycles, it drops 12%. Finally, after 2500 cycles, the voltage remains stable.

### 2.3. Finite Element Modeling

TENG devices have been simulated with COMSOL Multiphysics to compare the results of the electrical characterization with a theoretical model. The used module for this simulation is AC/DC Module (Electrostatics). The 2D model built in COMSOL consists of a box air (20 cm × 20 cm), the triboelectric material, and the copper electrodes, one is defined as ground and the other as floating potential. A specific surface charge density calculated by using Equation (1) has been assigned to the surface of the triboelectric material.

The model is created using a fine triangular mesh. The mesh contains more than 50,000 elements. Figure 7a shows the electric potential generated by the Mylar sample, with distance of 5 mm between the electrodes. Figure 7b,c show the output open-circuit voltage for different triboelectric materials without and with corona charging respectively. As seen, the simulated results are in agreement with the electrical measurements.

### 2.4. Fabrication of the Prototype

Figure 8a,b show the schematic and operating scheme of the TENG device based on contact separation mode. The specific working process in divided into three steps. In the first step, the spacers are flattered by the impact force so that the copper electrode fully contacts the dielectric film. Here, copper is used as the tribometallic material and top electrode. Due to the significant difference in the electronic affinity of the two materials, net positive charges are generated on the surface of the copper layer, and equal net negative charges are left on the surface of the dielectric film. Second, the spacers return to their original folded state due to their inherent elastic force, until the maximum vertical distance is reached. This conducts free electrons from the bottom copper electrode of the dielectric film to the top copper electrode through an external circuit to compensate for the potential difference between the two electrodes. Finally, the spacers are pressed again under another external mechanical impact, the top electrode and dielectric film are fully in contact again, and free electrons will flow from the top electrode through the external charge back to the bottom electrode. The previous three steps of the TENG formed a complete and repeatable cycle of electrical power generation.

For the manufacturing of the TENG device, a 70 mm × 70 mm × 0.5 mm PET (polyethylene terephthalate) substrate has been used for the bottom part. On the PET substrate, a piece of adhesive copper of 45 mm × 45 mm × 0.05 mm is adhered. This electrode will be smaller than the dielectric material to avoid possible shortcuts between electrodes. Finally, a Mylar layer is adhered to the bottom copper electrode, 50 mm × 50 mm × 0.05 mm. For the top electrode, we use a PET sample of the same dimensions as the bottom one, but with the different spacers engraved by laser cutting at the ends of the structure. A 45 mm × 45 mm × 0.05 mm piece of adhesive copper is also attached as the top electrode.

To improve the energy generated by the TENG device, two of them have been manufactured in a stack (Figure 8c,d) and have been measured individually, connected in series and in parallel. For this stack manufacturing, the process is identical, but, in this case, the top PET substrate of the first TENG and the bottom one of the second are shared to minimize cost and volume.

### 2.5. Electrical Characterization

For the electrical characterization of this TENG device, as shown in Figure 8, two different measurements have been carried out. First, we measured the maximum voltage generated by contacting the device directly to the sourcemeter. Second, we connected the device to a diode bridge and a capacitor of 10 µF to rectify the signal and store the energy. Then, by means of a switch, the stored energy is connected to 30 LEDs. For these measurements, in order to show a real application, instead of applying 50 N of force with a stepper motor, the force was applied with the palm of the hand, obtaining an average of 8 N of force.

For the first measurement, each of the dielectrics was measured separately. As shown in Figure 9a, the maximum voltages are very similar, reaching a value of 170 V. By connecting the two materials in series, the voltage increases to 250 V. Ideally, the voltage should be doubled, but manually applying the force and not contacting each dielectric with its top electrode in phase do not reach this maximum voltage. An improvement of 50% improvement over a single individually connected device is obtained.

In the second characterization, as previously mentioned and shown in the schematic of Figure 9b, the device output is connected to a diode bridge, together with a capacitor of 10 µF, and is charged up to 5 V. Once the 5 V is reached, which corresponds to a stored capacitor energy of 125 µJ, there are a few seconds without charging, and then the LEDs are connected through the switch to illuminate them for 150 ms (Figure 9d). Figure 9c shows the comparative graph of charge and discharge of the 10 µF capacitor when powering the 30 LEDs and applying force with the palm of the hand at an approximate frequency of 6 Hz. It can be observed that when using a single device, the charging process takes 35 s, while when connecting two devices in series, this time is reduced to 23 s. Decreasing the charging time of the capacitor by 35%. In the same way, if the two devices are connected in parallel, the charging time is reduced to 12 s. This decreases the time by 65%, compared to the single device. From these results, we can infer that a power of 10 µW can be generated by two devices connected in parallel.

## 3. Discussion and Conclusions

In this article, the fabrication of a TENG device has been carried out, which can collect the mechanical energy of the force applied by tapping with the palm of the hand. This energy could be used to power low-power electronic sensors in a self-sufficient way.

In order to manufacture the TENG devices, the most suitable materials (i.e., those with the highest surface charge transfer and most cost-effective) have been studied. In addition, the design of the spacers has been optimized to improve the contact surface and reduce the final costs of the device.

To increase the surface charge density, charge injection was carried out by means of high-voltage corona charging. The voltages to be applied and the optimal time for this charge injection were examined.

To validate the correct operation of the TENG device, electrical characterizations with different mounting configurations were carried out. By using a stack of only two devices connected in parallel, 30 LEDs were illuminated every 12 s, thanks to a power generation of 10 µW produced by hand tapping.

## Figures and Tables

**Figure 1 nanomaterials-12-03271-f001:**
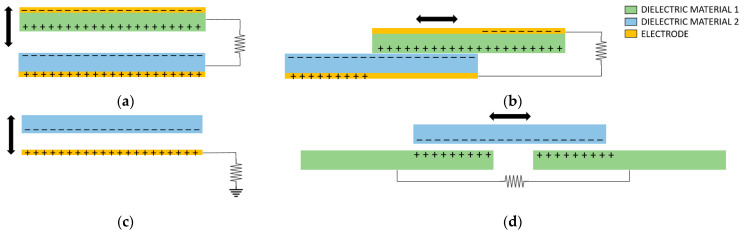
Four basic modes of work of a TENG: (**a**) vertical contact-separation mode; (**b**) lateral-sliding mode; (**c**) single-electrode mode; (**d**) free-standing mode.

**Figure 2 nanomaterials-12-03271-f002:**
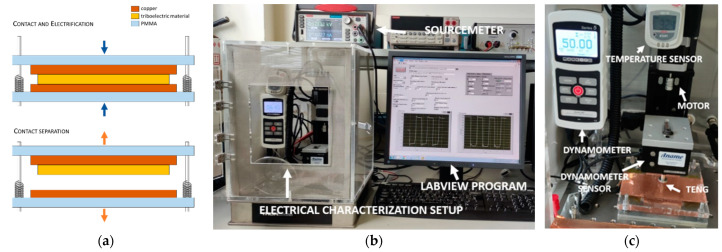
(**a**) Schematic of material and electrode used for the characterization of the materials; (**b**,**c**) ad hoc setup for electromechanical characterization of triboelectric devices.

**Figure 3 nanomaterials-12-03271-f003:**
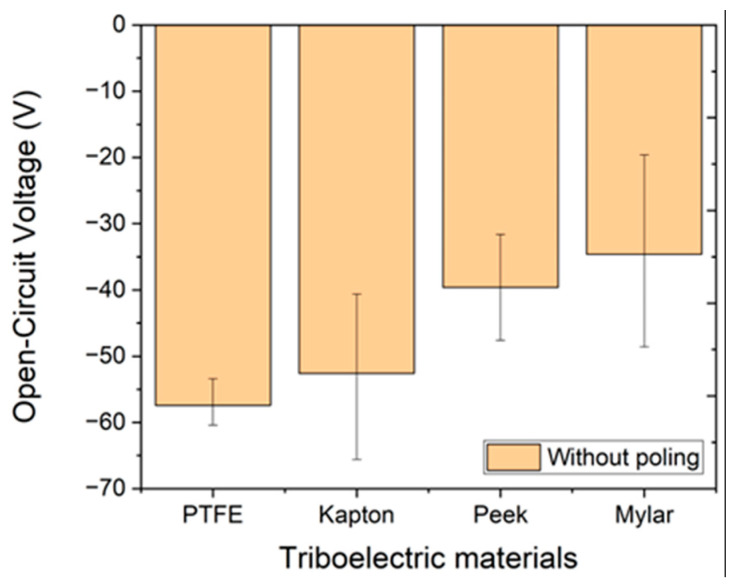
Comparison of voltages generated for the different material samples.

**Figure 4 nanomaterials-12-03271-f004:**
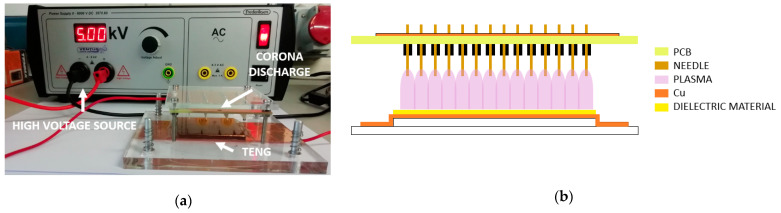
(**a**) Charge injection setup using high voltage corona charging; (**b**) schematic of the process of charge injection by corona.

**Figure 5 nanomaterials-12-03271-f005:**
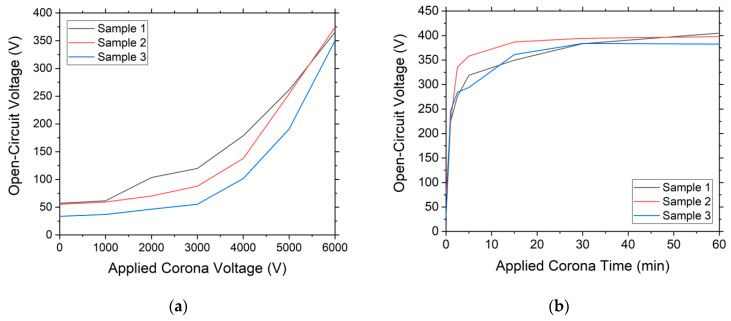
(**a**) Graph of the generated open-circuit voltage as a function of the voltage applied to the corona discharge; (**b**) graph of the open-circuit voltage that can be generated by the material as a function of the injection time with a constant corona discharge voltage of 6000 V.

**Figure 6 nanomaterials-12-03271-f006:**
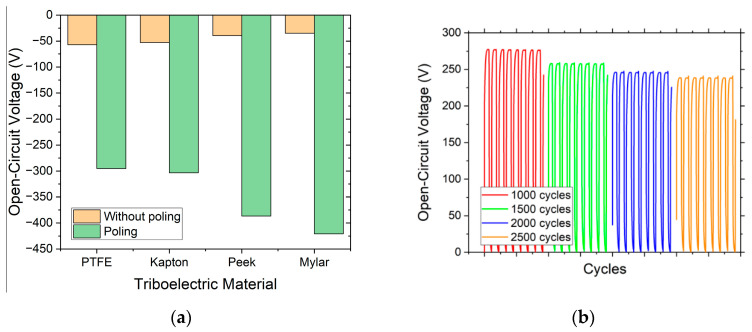
(**a**) Comparison of voltages generated with the different samples of polarized materials for 15 min and a voltage of 6000 V; (**b**) measurements of eight contact cycles for each material to validate the permanent effect of polarization.

**Figure 7 nanomaterials-12-03271-f007:**
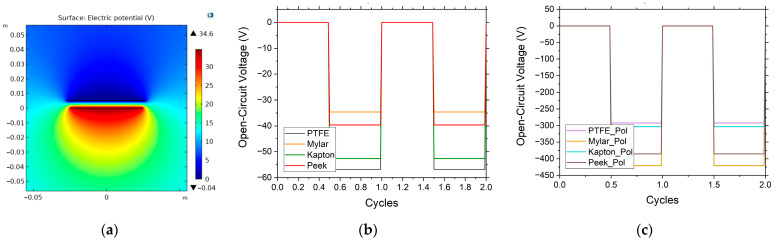
(**a**) Electric potential (in volts) generated by the Mylar sample with an electrode separation of 5 mm. (**b**,**c**) Open-circuit voltage for different triboelectric materials without and with corona charging, respectively.

**Figure 8 nanomaterials-12-03271-f008:**
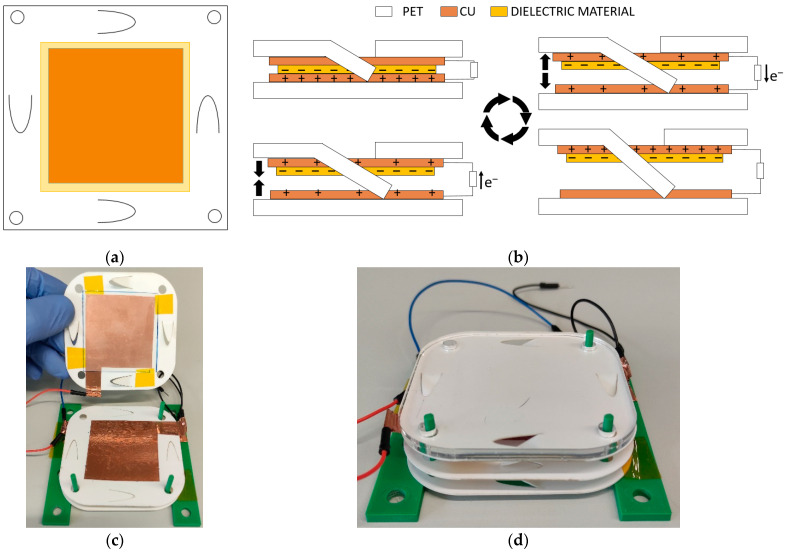
(**a**) Scheme of the TENG device; (**b**) scheme of operation of the TENG device; (**c**,**d**) image of the manufactured TENG device before and after assembling, respectively.

**Figure 9 nanomaterials-12-03271-f009:**
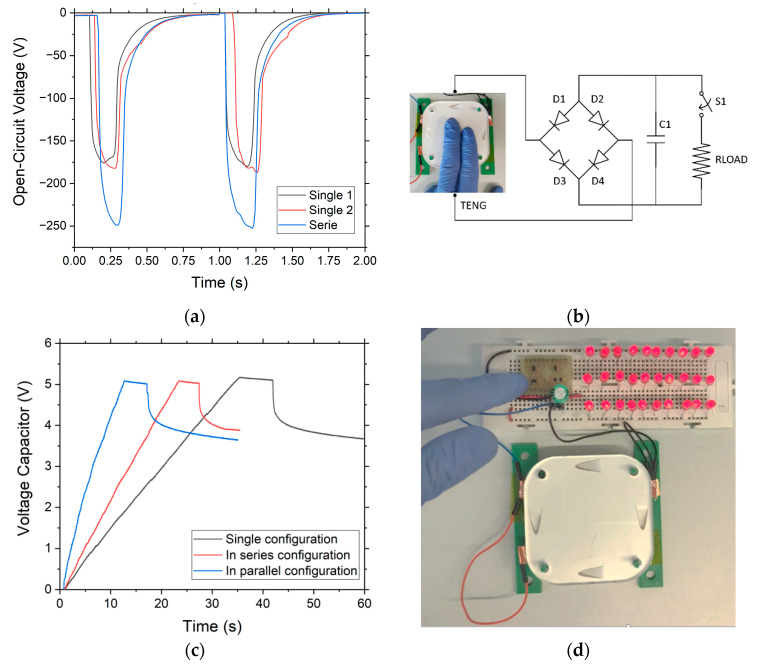
(**a**) Voltage generated by each device individually and connected in series; (**b**) circuit diagram and photograph of the TENG device powering the 30 LEDs; (**c**) process of charging and discharging the capacitor for a capacity of 10 µF; (**d**) TENG device used to light 30 LEDs up.

## Data Availability

Data are contained within the article or Appendix A.

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
