# Peer review of "Tapping-Actuated Triboelectric Nanogenerator with Surface Charge Density Optimization for Human Motion Energy Harvesting"

_nanomaterials, 2022, doi:10.3390/nano12193271_

Round 1

Reviewer 1 Report

The prototype device presented in the paper is very interesting.

However, there is no adequate theoretical modelling for validation.

The paper only reports experimental results, and there is no guidelines for the design and dimensioning of the prototype.

A strong theoretical section with experimental validation is required for a scientific paper.

The outcome of the paper cannot be thus appreciated and evaluated with confidence

Author Response

Reviewer 1

 The prototype device presented in the paper is very interesting.

However, there is no adequate theoretical modelling for validation.

The paper only reports experimental results, and there is no guidelines for the design and dimensioning of the prototype.

A strong theoretical section with experimental validation is required for a scientific paper. The outcome of the paper cannot be thus appreciated and evaluated with confidence.

We want to thank the reviewer the kind comments on our article. We agree that a theoretical modeling could improve the quality of our article.

Following the reviewer's recommendation, important revisions and an extensive editing of English language have been made in the manuscript. We have added some FEM simulations performed by COMSOL Multiphysics to improve the quality of our article, used to predict the behavior of the proposed device. From these results, we can infer the electrical behavior of our device. We have added this simulation as the new Fig 7a-c and it has been commented on the text.

Experimental details about the corona charge process have been added and description of the operating mechanism of the TENG device.

New figures (Figures 4b, 6b, 7a-c and 8b) and explanatory text have been added with new information. Furthermore, to make the article more understandable, several figures have been modified (Figures 2a, 3a and 6a) and commented in a clearer way.

Reviewer 2 Report

Article „Tapping-actuated triboelectric nanogenerator with surface charge density optimization for human motion energy harvesting” describes a TENG device using the triboelectric effect to convert mechanical to electrical energy with the description of additional optimization steps leading to better performance. The article is generally written OK; some of the issues that must be corrected/explained in the text is mentioned below. After the authors correct all of them, the article should be ready for publishing.

In lines 62 to 67, please specify how mentioned changes will improve the performance. Each time you cite, add e.g. and provide additional info so that readers do not have to go to the references immediately.

Line 95 what is the size and area of the stamp acting on the sample?

Figure 2a Where are the springs retracting the covers after touching, or a different separation technique is used?

Please describe more exact the corona discharge. Is the 5kV voltage applied to both electrodes touching the dielectric? Is there a current flow between electrodes, and if so, how much is it? Is this effect permanent?

Figure 7b What is this bend, middle PET part going from the top electrode?

Figure 8c Serie - change to in series. The second point is why the voltage got cut off at the 5V limit. Did you use the Zener diode in the setup that is not listed?

Text corrections:

Line 13  „TENG nanogenerator” Authors wrote “tortriboelectric nanogenerator nanogenerator” one is redundant. Provide also full name first time using an abbreviation.

Line 84 missing n „…chose,…”

Line 191 “…the mechanical energy of the force applied by tapping with the palm of the hand and could be used to power low-power electronic sensors in a self-sufficient and sustainable…”  Correct the sentence or do two out of it.

Author Response

Please attached our comments to the reviewer's report.

Reviewer 3 Report

In this work, the authors reported on different approaches to optimize energy generation through the correct choice of materials, and designed new spacers to increase the contact surface area and charge density of dielectric materials. A triboelectric nanogenerator was fabricated capable of harvesting the mechanical energy generated by hand-knocking force and powering small electronic sensors in a self-sufficient and sustainable manner. I suggest its publication after the following revisions.

1. In Figure 3, the ability of different dielectric materials to generate voltage is characterized, whether the test is carried out under the same environmental conditions, and the influence of environmental factors such as temperature and humidity is need to be considered.

2. In Figure 5, the optimal voltage and time of corona application were explored. The corresponding experimental details are needed to be provided, and the differences in these obtained curves should be discussed in detail.

3. Some related references about TENGs in human motion energy harvesting need to be also cited like Adv. Mater. Technol. 2022, 7, 2100702; Nano Energy, 2021, 86, 106058; ACS Nano, 2022, 16, 5909ï¼›Adv. Funct. Mater. 2021, 31, 2100940.

4. The authors should further describe the operating mechanism of the TENG device in Fig. 7b.

Author Response

Please find attached our comments to the reviewer's report.

Round 2

Reviewer 1 Report

This paper presents a prototype of tapping-actuated energy harvester for human motion.

The paper is very interesting and reads relatively well.

The following points should be addressed to improve the paper:

1.

The introduction is very poor and limited. It does not comprehensively introduce the topic to the potential reader.

2.

In the introduction reference is made to several applications of kinetic energy harvesting, for example human motion, waves, wind, water. Some reference to induced vibration by moving vehicles should be also included to broaden the reader view on the topic. For example, in:

- Dominant frequencies of train-induced vibrations, J. Sound Vib. 319 (2009) 247–259.

- Energy harvesting from the vibrations of a passing train: Effect of speed variability, Journal of Physics: Conference Series 744(1),012080, 2016

a perspective view is given to the dominant frequencies induced by passing trains for energy harvesting purposes.

3.

Other than tapping-induced vibration, intermitted, time-limited, and random excitations are available. These different forms of kinetic sources and devices should be introduced to the reader for a broader view on the topic. Those are covered, for example, in:

- Challenges for Energy Harvesting Systems Under Intermittent Excitation, IEEE J. Emerging Sel. Top. Circuits Syst., 4(3), pp. 364–374, 2014

- Harvesting Energy From Time-Limited Harmonic Vibrations: Mechanical Considerations, Journal of Vibration and Acoustics, 2017, 139, 051019

- A robust hybrid generator for harvesting vehicle suspension vibration energy from random road excitation, Applied Energy 309,118506, 2022,

4.

The paper presents an experimental work only. This should be clarified in the title, as for example “Tapping-actuated triboelectric nanogenerator with surface charge density optimization for human motion energy harvesting: an experimental work”, and well emphasized in the abstract as well

5.

It is stated that the device is used for human motion, but the supplementary video shows a relatively fast tapping motion. Which is the effect of tapping frequency on the performance of the device? What would happen if the tapping frequency was that typical of a walking person?

Author Response

The following points should be addressed to improve the paper:

  1. The introduction is very poor and limited. It does not comprehensively introduce the topic to the potential reader

The section “Triboelectric Generator” has been attached to the Introduction section, therefore this section is now large enough. In addition, the initial introduction has been improved by explaining the reason or need for the TENG device and the concept of IoT.

Moreover, the results achieved in this work has been more explicitly detailed at the end of the introduction. 

  1. In the introduction reference is made to several applications of kinetic energy harvesting, for example human motion, waves, wind, water. Some reference to induced vibration by moving vehicles should be also included to broaden the reader view on the topic. For example, in: 
  • Dominant frequencies of train-induced vibrations, J. Sound Vib. 319 (2009) 247–259.
  • Energy harvesting from the vibrations of a passing train: Effect of speed variability, Journal of Physics: Conference Series 744(1),012080, 2016

A perspective view is given to the dominant frequencies induced by passing trains for energy harvesting purposes.

Many thanks for the recommendations of references related to TENGs for harvesting energy from vibrations by moving vehicles. All the references have been added to the article. 

  1. Other than tapping-induced vibration, intermitted, time-limited, and random excitations are available. These different forms of kinetic sources and devices should be introduced to the reader for a broader view on the topic. Those are covered, for example, in: 
  • Challenges for Energy Harvesting Systems Under Intermittent Excitation, IEEE J. Emerging Sel. Top. Circuits Syst., 4(3), pp. 364–374, 2014
  • Harvesting Energy From Time-Limited Harmonic Vibrations: Mechanical Considerations, Journal of Vibration and Acoustics, 2017, 139, 051019
  • A robust hybrid generator for harvesting vehicle suspension vibration energy from random road excitation, Applied Energy 309,118506, 2022,

We have added the references suggested by the reviewer. They make broader the view on the topic to the reader. 

  1. The paper presents an experimental work only. This should be clarified in the title, as for example “Tapping-actuated triboelectric nanogenerator with surface charge density optimization for human motion energy harvesting: an experimental work”, and well emphasized in the abstract as well.

      We kindly thank the suggestion from the reviewer. However, we think that the title is now large enough and the content of the article has been now improved, showing theoretical concepts, simulations, as well as experimental results. Therefore, we believe that there is no need to specify this point in the title.

We have added the following sentence to the abstract to make clear the argument pointed out by the reviewer:

“This work shows the theoretical concept and simulations of the proposed TENG device, as well as the experimental work carried out.” 

  1. It is stated that the device is used for human motion, but the supplementary video shows a relatively fast tapping motion. Which is the effect of tapping frequency on the performance of the device? What would happen if the tapping frequency was that typical of a walking person? 

By reducing the frequency by 66% from approximately 6 Hz to 2 Hz, the charging time of the capacitor is increased by 130%. In the case of a person walking, the frequency could be around 2 Hz. The force generated on the TENG would increase from 8 N to 500 N - 1000 N and surely the power generated would be even greater than that shown in this article. However, the manufactured spacers are not designed to stand for such large forces. A reinforced version should be manufactured for walking applications and published as a new article.

In the load injection section of this article, it can be seen how by applying a force of 50 N and the same size of the triboelectric material, a voltage of 420 V is achieved, while with the force of 8 N exerted by the hand-tapping it is reduced to 180 V due to a smaller contact area. 

Reviewer 2 Report

In a detailed fashion, the authors answered all the questions raised before. All the proposed corrections were implemented in the text. The article improved enormously. I see no more issues hindering publication.

Author Response

We would like to thank the reviewer for the kind comments on our article. thanks to the reviewer's corrections, the article has been greatly improved.

Round 3

Reviewer 1 Report

The authors have revised the paper according to the reviewer's comments.

The paper has been improved.

One minor note is about the ref. [27] in the reference list, which should be formatted and corrected appropriately.